# Clinical and Epidemiological Characteristics of Chikungunya and Dengue Infections in Provincial Hospitals of Davao de Oro, Philippines

**DOI:** 10.3390/reports7040112

**Published:** 2024-12-11

**Authors:** Nestor Arce, Kobporn Boonnak, Lee Thunder Bernasor, Christian Joy Salas, Anastasia Putri, Pyae Linn Aung, Hisham Ahmed Imad, Wirongrong Chierakul, Viravarn Luvira, Benjaluck Phonrat, Weerapong Phumratanaprapin, Prakaykaew Charunwatthana

**Affiliations:** 1Department of Clinical Tropical Medicine, Faculty of Tropical Medicine, Mahidol University, Bangkok 10400, Thailand; nestor.arce@jmc.edu.ph (N.A.J.); anastasia.put@student.mahidol.ac.th (A.P.); hishamahmed.ima@mahidol.ac.th (H.A.I.); wirongrong.chi@mahidol.ac.th (W.C.); viravarn.luv@mahidol.ac.th (V.L.); benjaluck.pho@mahidol.ac.th (B.P.); weerapong.phu@mahidol.ac.th (W.P.); 2Jose Maria College Foundation, Inc., College of Medicine, Davao City 8000, Philippines; ltbernasor@gmail.com; 3Department of Immunology, Faculty of Medicine, Siriraj Hospital, Mahidol University, Bangkok 10700, Thailand; kobporn.boo@mahidol.ac.th; 4College of Nursing, University of the Immaculate Conception, Davao City 8000, Philippines; csalas@uic.edu.ph; 5Mahidol Vivax Research Unit, Faculty of Tropical Medicine, Mahidol University, Bangkok 10400, Thailand; pyaelinn.aun@mahidol.ac.th; 6Center for Infectious Diseases Education and Research, Department of Viral Infections, Research Institute for Microbial Diseases, Osaka University, Osaka 565-0871, Japan; 7Mahidol Oxford Tropical Medicine Research Unit, Faculty of Tropical Medicine, Mahidol University, Bangkok 10400, Thailand

**Keywords:** acute febrile illness, chikungunya, dengue, coinfection, Southeast Asia, Philippines

## Abstract

**Background:** Mosquito-borne diseases such as dengue and chikungunya are endemic to tropical regions and are common causes of acute febrile illness in both adults and children. The Philippines, home to more than one hundred million residents and visited by several million tourists each year, is one such region where the risk of these diseases is endemic. **Objective and Methods:** To better understand the detailed situation, we estimated the proportion of these diseases in the community by conducting a prospective observational study in four provincial hospitals of Davao de Oro, Philippines, from February 2019 to February 2020. Sera from 382 study participants were used for laboratory confirmation of dengue or chikungunya by serology. **Results:** Dengue was diagnosed in 57.1%, chikungunya in 7.07%, and coinfection with both dengue and chikungunya in 7.3%, while the etiology was undetermined in 35.9% of the study participants. Common clinical symptoms included fever, headache, and a rash, which were overlapping symptoms that were clinically indistinguishable at the presentation to the hospital, necessitating laboratory diagnostics. **Conclusions:** The identification of the presence of chikungunya in Davao de Oro calls for increased awareness, improved diagnostics, and integrated disease control measures to manage outbreaks that can occur in dengue-endemic regions.

## 1. Introduction

Mosquito-borne viruses such as dengue and chikungunya are frequent causes of acute febrile illnesses in Southeast Asia [1]. Among these two viruses, dengue in Southeast Asia accounts for 70% of its burden in tropical regions of Asia, with Southeast and South Asia experiencing the highest numbers of cases and deaths, reflecting the significant and growing impact of arboviral infections in this region [2]. Dengue viruses belong to the *Orthoflavivirus* genus and consist of four distinct serotypes (DENV 1–4), each containing multiple genotypes and offering transient cross-protective immunity [3]. Chikungunya virus is an Alphavirus virus with three genotypes: the West African lineage, East, Central and South African (ECSA) lineage, and Asian lineage [4]. During the initial presentation at the hospital, distinguishing between the two diseases can be challenging, as they manifest overlapping symptoms. Laboratory tests, such as antigen or antibody assays or PCR, are necessary for a definitive diagnosis [5].

However, the clinical trajectories of both diseases differ greatly. In dengue, most patients remain asymptomatic, and those who develop symptoms typically experience a self-limited illness. In some cases, symptoms may include bleeding, hepatitis, plasma leakage, and shock. On the other hand, in chikungunya, most patients develop symptoms characterized by arthralgia and arthritis. The clinical course is also much longer, with symptoms of arthritis persisting for several months.

In the Philippines, cases of chikungunya and dengue have been reported since the late 1950s. Dengue remains a major public health concern, leading the government to provide free diagnostics under the national insurance scheme, enabling large-scale screening of febrile patients [6]. The Philippines was also the first country in Asia to approve and offer a dengue vaccine, reflecting the disease’s significant burden.

In contrast, chikungunya has received less public health attention, likely due to its lower mortality rates compared to dengue. However, chikungunya causes severe long-term effects, including reduced productivity and loss of workdays [7,8]. A cohort study in Cebu reported three symptomatic chikungunya infections per 100,000 person-years during a three-year outbreak [9].

The Philippines, which is composed of 81 provinces, including Davao de Oro, comprises 11 municipalities, has a population of approximately 767,547 people, and features four secondary provincial hospitals located in Montevista, Laak, Pantukan, and Maragusan. Davao de Oro is home to migrants from Luzon and Visayas, as well as ethnic tribes such as the Mansaka, Mandaya, Davaoeno, and Kalagan. The main sources of income in the area are business establishments, banana plantations, and vast silver and gold mines across the province.

Chikungunya is likely underreported in this area for several reasons. Unlike dengue, for which free diagnostic tests are provided by the government to registered Filipinos under national insurance schemes, diagnostic tests for chikungunya are not widely available. Moreover, many residents, particularly migrants and their families working in the agricultural and mining sectors of Davao de Oro, may not be registered in the national insurance scheme and therefore lack access to diagnostics. This gap in access to diagnostics could contribute to the underdiagnosis and underreporting of chikungunya in this region. Understanding the true prevalence of chikungunya in this context is crucial for implementing effective public health strategies and improving healthcare access for all residents of Davao de Oro. In this study, we aimed to estimate the proportion of acute chikungunya and dengue infections among patients with acute febrile illness in four provincial hospitals in Davao de Oro in the Davao region.

## 2. Materials and Methods

This multicenter, prospective study was conducted at four provincial hospitals in Davao de Oro Province, located in the southeastern part of the Davao region. The hospitals included in the study were Montevista Provincial Hospital, Laak Provincial Hospital, Pantukan Provincial Hospital, and Maragusan Provincial Hospital. Patients who presented to the emergency department with complaints of fever or who had experienced at least one episode of fever of 38.5 °C or higher within the past week, and who were suspected of having dengue infection or had a positive result for dengue serology, were recruited for this study. Serum was used for the subsequent analysis of the chikungunya serology.

### 2.1. Dengue NS1 Antigen and IgM/IgG Antibody Assay

A lateral-flow, rapid point-of-care diagnostic test kit for dengue, manufactured by Chembio Diagnostics, Inc., Medford, NY, USA) was used to detect the dengue NS1 antigen and anti-dengue virus IgM and IgG antibodies. The tests were performed according to the manufacturer’s instructions, using 10 μL of serum. The results were interpreted 15 min after the test was conducted. The test was considered positive if two bands appeared, indicating the presence of the NS1 antigen and IgM, along with the control band. If no bands appeared, the test was considered negative. The appearance of IgG alone compared with the control band indicated a previous infection. If the control band failed to appear, the test was deemed invalid.

### 2.2. Chikungunya IgM Antibody Assay

To evaluate the presence of anti-chikungunya virus IgM antibodies in this study, we utilized a commercially available indirect enzyme-linked immunosorbent assay (EUROIMMUN, Lübeck, Germany). The assay was performed in accordance with the manufacturer’s instructions. In brief, 10 μL of serum was diluted with the buffer solution provided by the manufacturer and added to a precoated 96-well microplate with a serum dilution of 1:100. Next, the microplate underwent three wash cycles with working-strength wash buffer, and the antibodies were identified with the addition of an enzyme conjugate (peroxidase-labeled anti-human IgM). A substrate solution containing THM/H_2_O_2_ was subsequently added to induce the development of a blue color. The process was halted by the addition of 0.5 M sulfuric acid, and the photometric measurement was recorded at a wavelength of 450 nm. The optical density was used to quantify the concentration of specific anti-chikungunya virus antibodies present in the serum samples. The results were evaluated through a semiquantitative approach, calculating the ratio of the extinction of the patient serum sample to the extinction of the calibrator. This ratio was interpreted as follows: <0.8, negative; ≥0.8 to <1.1, borderline; and ≥1.1, positive.

### 2.3. Case Definition

Diagnoses were considered confirmed if they met any of the following criteria: Dengue infection (DENV) was classified as positive (DENV +ve) if the sample tested positive for NS1 antigen and/or IgM antibodies in serological testing. Chikungunya infection was considered positive (CHIKV +ve) if serological testing revealed IgM antibodies specific to chikungunya. A mixed infection (CHIKV +ve and DENV +ve) was diagnosed when the specimen tested positive for chikungunya IgM antibodies along with dengue markers, either NS1 antigen and/or IgM antibodies. Finally, cases were classified as unknown (CHIKV −ve and DENV −ve) if all the tests mentioned above were negative, showing no laboratory evidence that attributed the illness to either dengue or chikungunya.

### 2.4. Statistical Analysis

One-way ANOVA was used to assess parametric variables across multiple groups, such as days of fever and hematocrit levels, whereas the Mann—Whitney U test was employed to evaluate differences in nonparametric variables, such as age, WBC count, and platelet count, between groups. The chi-square test was used to assess categorical variables, and Fisher’s exact test was applied specifically when more than 20% of the cells had expected frequencies less than 5. A *p*-value of less than 0.05 was considered significant for all tests performed. All the statistical analyses were conducted via Microsoft Excel and STATA version 17 for MacOS 14.5 (23F79).

## 3. Results

Over a 14-month study period, 382 patients were screened for chikungunya and dengue infection at four provincial hospitals in Davao de Oro, Philippines. Among the patients identified as having acute febrile undifferentiated illness, 27 (7.07%) tested positive for circulating anti-chikungunya IgM, and 190 (49.74%) tested positive for dengue infection, with the detection of both the dengue NS1 antigen and the anti-dengue IgM antibody. Additionally, 28 (7.33%) patients were identified as coinfected with both chikungunya virus and dengue virus, as their serum tested positive for anti-chikungunya IgM and positive for dengue NS1 and anti-dengue IgM. The remaining 137 patients (35.86%) had undetermined infectious etiologies, as shown in Figure 1.

### 3.1. Demographic and Epidemiological Characteristics

The age distribution varied significantly between the two diseases (*p* = 0.057), with median ages of 12 years for chikungunya and 17.5 years for dengue. Similarly, the median age was 17 years for patients identified as coinfected and for those who tested negative for both dengue and chikungunya, where the infective etiology remained unknown. The sex distribution of males: females across infection groups was as follows: chikungunya (9:18), dengue (103:87), coinfection (14:14), and unknown etiology (80:57). However, there appeared to be a greater proportion of females with chikungunya than with dengue.

### 3.2. Hematological Profile of the Study Cohort

The mean hematocrit levels were similar across disease groups, ranging from 40.7% to 43.3%. The leukocyte count was observed to have a statistically significant difference among the groups (*p* = 0.0046). The median leukocyte count was lowest in patients with coinfections (3.3 × 10^3^ per microliter), with a range of 1.3–8.7 × 10^3^/µL. In patients with dengue infection, the mean leukocyte count was 3.7 × 10^3^/all, and in patients with chikungunya infection, it was 4.05 × 10^3^/all. In the group in which the etiology was undetermined, the mean leukocyte count was 4.5 × 10^3^/μL. The median platelet counts were lowest in dengue infections (119 × 10^9^/L, range: 30–395), followed by coinfections (122 × 10^9^/L, range: 40–255) and chikungunya infections (144 × 10^9^/L, range: 90–332).

### 3.3. Clinical Manifestations of Chikungunya, Dengue, and Febrile Patients with Undetermined Etiology

Fever was the most prevalent symptom across all disease groups, affecting all patients. The mean duration of fever was also similar, ranging from 3.5 to 3.7 days. Headache was observed to be another predominant symptom, commonly reported by 66.7% of patients with chikungunya virus infection, 73.2% of dengue patients, and 75% of patients with coinfections. Retroorbital pain, which is often associated with dengue fever, was not significantly different among the groups, ranging from 7.9% in dengue patients to 14.8% in chikungunya patients.

Musculoskeletal symptoms varied among the disease groups. Arthralgia was more prevalent in chikungunya (44.4%) and coinfections (42.9%) than in dengue (32.6%). However, the reporting of myalgia was similar across disease groups (32.1–37.0%). A rash was more common in chikungunya (59.3%) and dengue (63.7%) infections than in coinfections (28.6%) (*p* = 0.052).

Gastrointestinal symptoms varied among the groups. Nausea was more common in the group with coinfection (78.6%), and vomiting followed a similar pattern: patients with coinfections had more vomiting (64.3%), but vomiting was also common in dengue (53.7%) and chikungunya (44.4%). Abdominal pain was more commonly reported in patients with dengue (61.6%)

Neurological involvement, represented by confusion, was rare across all groups. However, it was slightly more prevalent in chikungunya (3.7%) and coinfections (3.6%) than in dengue (0.5%).

### 3.4. Distribution of Patients Who Presented to Provincial Hospitals

The distributions of cases at the four provincial hospitals significantly differed (*p* = 0.018). Laak Provincial Hospital reported a relatively even distribution of infection cases across types, with 37% chikungunya cases, 29.5% dengue cases, 28.6% coinfections, and 29.2% unknown cases. Pantukan Provincial Hospital reported 37.0% chikungunya cases, 27.4% dengue cases, a higher rate of coinfections at 46.4%, and 25.6% unknown cases. Maragusan Provincial Hospital had a notably lower proportion of chikungunya cases (14.8%) than dengue cases (3.2%) and no coinfections. In contrast, Montevista Provincial Hospital had a greater proportion of dengue cases (40%) than it did chikungunya (11.1%) and coinfections (25%), as shown in Table 1.

## 4. Discussion

### 4.1. Proportions of Patients with Chikungunya and Dengue in the Study Cohort

Our results demonstrate that among patients presenting with acute febrile illness in Davao de Oro, dengue remains the predominant arboviral infection, accounting for 49.74% of cases identified in this cohort. However, the detection of chikungunya in 7.07% of the cases, along with 7.33% coinfections, reflects the emergence of chikungunya virus as a significant public health concern in the region. This has previously not been reported in the region of Davao de Oro, as in other areas of the Philippines [10]. These findings from Davao de Oro suggest that chikungunya virus may be more widespread than is currently recognized.

Furthermore, coinfections present a unique challenge in terms of diagnosis, clinical management, and potential disease severity. The global pooled prevalence of dengue and chikungunya coinfection has been reported to be 2.5%, with 95% CI: 1.8–3.4 [11]. To our knowledge, prior studies from the Philippines did not detect any coinfections [10]. Nevertheless, Asia, as a region, has the highest coinfection prevalence, at 3.3% (95% CI: 2.3–4.6) [11]. Our results suggest that coinfected patients may exhibit overlapping symptoms characteristic of both viral infections, potentially complicating clinical assessment and management.

However, in one-third of the patients (35.86%), neither chikungunya nor dengue was found, and the infective etiology remained undetermined. We postulate that this group of patients might have had other Orthoflaviviruses, such as Zika virus and Japanese encephalitis virus [12,13]. As these viruses remain endemic in the region, they should be considered in the differential diagnosis of arboviral infection in Southeast Asia.

Among other viral etiologies to consider, those stratified according to age groups—children, adolescents, and adults—are DNA viruses such as human herpesviruses 4 and 5, which can present as mononucleosis, and HHV6, HHV7, or parvovirus B19, which can manifest as erythema infectiosum [14]. Children and adolescents may be at risk of these infections. Other RNA viruses to consider in patients with febrile exanthema and enanthema include measles, rubella, and HIV [15,16,17]. If there has been exposure to insects such as ticks or sick domestic animals and all the above possibilities have been ruled out, we also suggest screening for severe fever with thrombocytopenia syndrome virus (SFTSV) or bacterial illnesses such as rickettsiosis, leptospirosis, and salmonellosis. This plethora of pathogens reflects the complexity of diagnosing acute febrile illnesses in tropical regions, emphasizing the need for broader diagnostic capabilities to identify other potential pathogens causing similar clinical presentations.

### 4.2. Clinical Manifestations and Hematological Profiles of Chikungunya and Dengue

The overlap in clinical symptoms between chikungunya and dengue infections poses a significant challenge for differential diagnosis on the basis of clinical findings at presentation alone. The higher prevalence of arthralgia in chikungunya (44.4%) and coinfected patients (42.9%) than in dengue patients (32.6%) aligns with the characteristic joint pain associated with acute chikungunya infection. This finding highlights the importance of considering chikungunya in patients who present with acute febrile illness with prominent joint pain, especially with articular tenderness and peripheral joint involvement. Arthralgia can present with other viruses described above. For example, rubella infection in adults can involve polyarthralgia of large joints, and similar clinical manifestations or arthropathy have also been reported in adults with parvovirus B19 [18,19].

Interestingly, our study revealed a greater prevalence of rashes in both chikungunya (59.3%) and dengue (63.7%) patients than in coinfected patients (28.6%). A rash is a known feature of both infections, and the lower prevalence in coinfections was unexpected. Nevertheless, a febrile rash might be missed without a careful physical examination. In particular, blanchable rashes are associated with arboviral infections.

Dengue warning signs, such as abdominal pain, were observed more frequently in dengue cases (61.6%) than in chikungunya (37.0%) and coinfections (39.3%). In resource-limited settings where rapid dengue diagnostics are unavailable, a clinical diagnosis can be made by considering the epidemiology, symptoms, and warning signs [20]. Warning signs such as abdominal pain are very useful in monitoring the clinical trajectory of suspected dengue cases, allowing for effective patient triage and prompt management.

In addition to identifying warning signs, the hematological profile can be used to monitor the progression of the disease. Specifically, in dengue, the hematological profile reflects dynamic bone marrow suppression, resulting in leukopenia. The leukocyte count is among the first indicators to increase during recovery. When compared with chikungunya and cases of unknown etiology, significantly lower leukocyte counts are detected in dengue and coinfected patients, which is consistent with the known association of leukopenia with dengue infection.

The trend of lower platelet counts in dengue and coinfected patients, although not statistically significant, is consistent with the thrombocytopenia commonly observed in dengue patients. The relatively high platelet counts in chikungunya patients suggest that severe thrombocytopenia is less likely in patients with chikungunya infection, which could be a useful distinguishing feature in clinical settings.

### 4.3. Distribution of Chikungunya and Dengue in Provincial Hospitals of Davao de Oro

The significant differences in case distributions among the four provincial hospitals highlight the importance of considering local epidemiological patterns in arboviral disease surveillance and control. The higher proportions of chikungunya cases at Laak Provincial Hospital and Pantukan Provincial Hospital, for example, suggest a potential localized outbreak or environmental factors favoring chikungunya transmission in that area. Conversely, the higher proportion of dengue cases in Montevista Provincial Hospital indicates that dengue remains the dominant arboviral threat in some parts of the region. Furthermore, these findings suggest that chikungunya may be significantly underreported in Davao de Oro. The lack of routine diagnostic testing for chikungunya, combined with its clinical similarity to dengue, likely contribute to underdiagnosis and underreporting. This situation is exacerbated by the focus on dengue in national health programs, including the provision of free dengue diagnostics under the national insurance scheme [6].

The high proportion of cases with an unidentified etiology (35.86%) underscores the diagnostic challenges in resource-limited settings, impacting patient management and hindering disease surveillance and public health planning. The cocirculation of chikungunya and dengue viruses in Davao de Oro necessitates enhanced surveillance through comprehensive arboviral programs, improved access to diagnostic tests, and clinical education to distinguish between chikungunya and dengue.

We would like to acknowledge a limitation in this study: the diagnosis was based primarily on clinical findings supported by serology. Without further investigations, such as virus isolation or molecular tests, we cannot rule out the possibility of cross-reaction among flaviviruses that are endemic in the region.

## 5. Conclusions

Dengue remains the predominant arboviral infection in Davao de Oro, affecting nearly half of patients with acute undifferentiated febrile illness. However, the emergence of chikungunya, previously underreported, indicates a broader circulation of arboviruses in the region. Variations in case distribution across hospitals reflect the need for localized surveillance and improved diagnostics. For clinicians, when faced with acute undifferentiated febrile illness in tropical regions, it is crucial to first exclude dengue using the available tests, provided other treatable tropical diseases are ruled out. If dengue is negative, clinicians should consider alternative etiologies such as chikungunya and Zika, which are epidemiologically present in the region.

## Figures and Tables

**Figure 1 reports-07-00112-f001:**
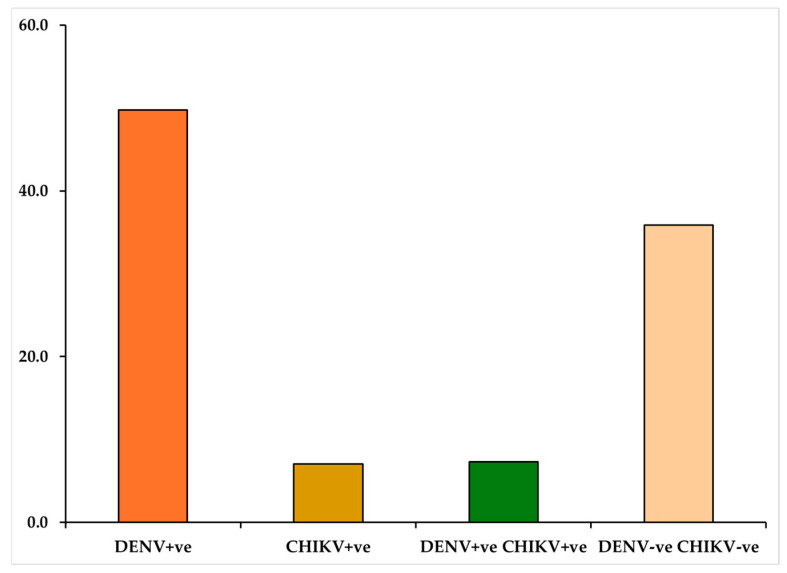
The proportions of positive serology for dengue and chikungunya among 382 patients with acute undifferentiated febrile illness.

**Table 1 reports-07-00112-t001:** Comparison of disease groups in the study cohort.

Characteristics (*n* = 382)	CHIKV +ve	DENV +ve	CHIKV +ve and DENV +ve	CHIKV −ve and DENV −ve	*p*-Value
Total tested (*n* (%))	27 (7.07)	190 (49.74)	28 (7.33)	137 (35.86)	
Age (median [min–max])	12 [1–49]	17.5 [0–65]	17 [0–43]	17 [0–65]	0.057
Sex (male: female)	9:18	103:87	14:14	80:57	0.117
Traveled	18%	15.2%	10.7%	15.3%	0.880
Hospital Area					
Laak Prov. Hosp.	37.0%	29.5%	28.6%	29.2%	0.018
Maragusan Prov. Hosp.	14.8%	3.2%	0%	6.6%	
Montevista Prov. Hosp.	11.1%	40%	25%	38.7%	
Pantukan Prov. Hosp.	37.0%	27.4%	46.4%	25.6%	
Clinical Manifestation					
Fever	100%	98.95%	92.86%	96.35	0.12
Days of Fever (Mean ± SD)	3.7 ± 1.3	3.7 ± 1.3	3.6 ± 1.7	3.5 ± 1.4	0.700
Headache	66.7%	73.2%	75%	67.1%	0.605
Nausea	51.9%	63.7%	78.6%	60.6%	0.194
Vomiting	44.4%	53.7%	64.3%	46%	0.224
Diarrhea	3.7%	10%	0	11%	0.228
Abdominal Pain	37.0%	61.6%	39.3%	37.2%	0.087
Skin rash	59.3%	63.7%	28.6%	49.6%	0.052
Retroorbital Pain	14.8%	7.9%	14.3%	10.9%	0.503
Arthralgia	44.4%	32.6%	42.9%	29.9%	0.334
Myalgia	37.0%	34.2%	32.1%	35.0%	0.982
Confusion	3.7%	0.5%	3.6%	0	0.056
Laboratory Profile at First-Day Presentation					
Hematocrit (mean ± SD)	40.7 ± 5.5	42.5 ± 5.7	41.0 ± 4.0	43.3 ± 5.4	0.058
WBC (median [min–mix])	4.05 [2.1–9.2]	3.7 [1.4–19.5]	3.3 [1.3–8.7]	4.5 [1.7–19.4]	0.004
Neutrophils (mean ± SD)	0.59 ± 0.19	0.52 ± 0.17	0.41 ± 0.22	0.51 ± 0.18	0.036
Lymphocytes (median [min–mix])	0.385 [0.07–0.99)	0.33 [0.07–0.8]	0.41 [0.1–0.82]	0.36 [0.05–0.78]	0.371
Platelets (median [min–mix])	144 [90–332]	119 [30–395]	122 [40–255]	126 [40–413]	0.051

Age is provided as the median with the maximum and minimum ranges, while all other continuous variables are provided as the mean ± SD. The *p*-value was determined to be <0.005. Chi-square analysis was used for categorical comparisons of symptoms, and the Mann—Whitney nonparametric test was used for continuous variables.

## Data Availability

The data presented in this study are available upon request from the corresponding author. The data are not publicly available to protect the privacy of the study participants.

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
