# Peer review of "Clinical and Epidemiological Characteristics of Chikungunya and Dengue Infections in Provincial Hospitals of Davao de Oro, Philippines"

_reports, 2024, doi:10.3390/reports7040112_

Round 1

Reviewer 1 Report

Comments and Suggestions for Authors

This study used medical and laboratory data of patients identified from four provincial hospitals of Davao de Oro, Philippines to calculate the proportion of dengue and chikungunya and describe the clinical characteristics of patients. Some limitations of the study should be noted.

In the Materials and Methods section, authors mentioned that only patients aged 1 to 65 years old who presented to the emergency department with fever and suspected dengue infection or a positive result for dengue serology were included study. Authors should justify –

-           Why infants below 1 year old were not included?

-           Why elderly patients aged over 65 years old were not included?

-           Why patients needed to be suspected with dengue infection or with a positive result for dengue serology? It’s possible that patients with chikungunya who were not suspected with dengue were not included in the study?

Author Response

Comment 1:

This study used medical and laboratory data of patients identified from four provincial hospitals of Davao de Oro, Philippines to calculate the proportion of dengue and chikungunya and describe the clinical characteristics of patients. Some limitations of the study should be noted.

In the Materials and Methods section, authors mentioned that only patients aged 1 to 65 years old who presented to the emergency department with fever and suspected dengue infection or a positive result for dengue serology were included study. Authors should justify –

Why infants below 1 year old were not included?

Why elderly patients aged over 65 years old were not included?

Why patients needed to be suspected with dengue infection or with a positive result for dengue serology? It’s possible that patients with chikungunya who were not suspected with dengue were not included in the study?

Response:

We apologize for the confusion caused by the inclusion of the specific age category in the Material and Methods section of the manuscript. The age range of 1 to 65 years was initially specified because the ethics committee classified individuals outside this range as potentially vulnerable, while our intention was to include all patients presenting with fever to the emergency department

Thank you for your comment. The study was designed to reflect the natural clinical scenario in an emergency room setting, where dengue is a high-burden disease in the Philippines. Given the prevalence of dengue, many clinicians prioritize screening for dengue virus in febrile patients. Unfortunately, this approach can sometimes lead to a bias, where other diseases such as chikungunya may not be initially considered, especially during non-outbreak periods. While chronic chikungunya cases may present to outpatient clinics with persistent arthritis and arthralgia, distinguishing acute dengue from acute chikungunya infection in an emergency setting can be challenging. In the absence of a detailed clinical history or specific clinical features that favor chikungunya, it is difficult to differentiate between the two diseases. Acute chikungunya infection can present similarly to acute dengue, and distinguishing them often requires a high level of clinical suspicion by practitioners experienced in managing arboviral diseases.

Reviewer 2 Report

Comments and Suggestions for Authors

Introduction

The introduction effectively contextualizes the relevance of dengue and chikungunya, highlighting their prevalence in the Philippines. It emphasizes the need for differential diagnosis due to the clinical similarities between the diseases, addressing a significant issue. I suggest including more recent global or regional statistical data to reinforce the scope of the problem. The connection between historical data and the study's motivation could be clearer and more concise. It would also be valuable to delve deeper into the study’s practical implications for local public health policies.

Methodology

The study provides detailed criteria for patient inclusion and exclusion, ensuring replicability. It employs standardized diagnostic methods (e.g., NS1 and IgM tests for dengue and ELISA for chikungunya). I suggest better justifying the choice of diagnostic tools and discussing associated limitations (e.g., sensitivity/specificity). The description of the study period could include context on seasonality, which is relevant for vector-borne diseases. Additionally, expanding details about team training or standardization across hospitals would help ensure data consistency.

Results

The study presents a detailed analysis of proportions for chikungunya, dengue, coinfections, and unknown etiologies. Data are well-presented in tables and graphs, aiding visual comprehension. I suggest further exploring the geographical distribution of infections and possible associated environmental factors. The section does not sufficiently address the impact of the findings (e.g., implications for hospital management). A deeper analysis of coinfections and their clinical implications would add significant value.

Discussion

The discussion interprets the findings in comparison to existing literature, strengthening the study's credibility. It highlights limitations, such as reliance on serology, and proposes alternatives for future research. However, some topics are treated too generically (e.g., alternative arboviruses for unknown etiologies). I suggest including more specific data or hypotheses. The discussion could also delve deeper into the impact of socioeconomic and cultural factors on the underreporting of chikungunya. Proposals for concrete interventions based on the findings, such as diagnostic or vector control strategies, would be beneficial.

Conclusion

The conclusion reaffirms the study's relevance, highlighting dengue’s predominance and the emergence of chikungunya as a public health issue. It emphasizes the need for local surveillance and diagnostic improvements. However, it could more explicitly propose immediate actions or guidelines for regional hospitals. The conclusion repeats some discussion points without adding new insights.

References

The references cover a good range of recent and relevant sources, addressing fundamental topics like diagnosis, clinical impact, and arbovirus epidemiology. I suggest including more local or regional sources to reinforce the geographical specificity of the study.

Author Response

Comment 1: 

The introduction effectively contextualizes the relevance of dengue and chikungunya, highlighting their prevalence in the Philippines. It emphasizes the need for differential diagnosis due to the clinical similarities between the diseases, addressing a significant issue. I suggest including more recent global or regional statistical data to reinforce the scope of the problem. The connection between historical data and the study's motivation could be clearer and more concise. It would also be valuable to delve deeper into the study’s practical implications for local public health policies.

Response: Thank you for your suggestion. In the revised manuscript, we have added a sentence to emphasize the global scope of the problem. Additionally, we have revised the third and fourth paragraphs of the Introduction to make the connection between historical data and the study’s motivation clearer and more concise

We have revised and added the following sentence to Lines 46 to 49

"Among these two viruses, dengue virus alone accounts for 70% of the global burden of dengue in tropical regions of Asia, with Southeast and South Asia experiencing the highest number of cases, deaths, reflecting the significant and growing impact of arboviral infections in this region"

We have made revision to 3rd and 4th paragraph in between Lines 63 to 72

"In the Philippines, cases of chikungunya and dengue have been reported since the late 1950s. Dengue has remains a major public health concern, leading the government to provide free diagnostics under the national insurance scheme, enabling large-scale screening of febrile patients (8). The Philippines was also the first country in Asia to approve and offer a dengue vaccine, reflecting the diseases significant burden.

In contrast, chikungunya has received less public health attention, likely due to its lower mortality rates compared to dengue. However, chikungunya causes severe long-term effects, including reduced productivity and loss of workdays (7, 9, 10). A cohort study in Cebu, reported 3 symptomatic chikungunya infections per 100,000 person-years during a three year outbreak (11)."

Comment 2: 

The study provides detailed criteria for patient inclusion and exclusion, ensuring replicability. It employs standardized diagnostic methods (e.g., NS1 and IgM tests for dengue and ELISA for chikungunya). I suggest better justifying the choice of diagnostic tools and discussing associated limitations (e.g., sensitivity/specificity). The description of the study period could include context on seasonality, which is relevant for vector-borne diseases. Additionally, expanding details about team training or standardization across hospitals would help ensure data consistency.

Response: 

The Chembio Diagnostics rapid test kit was used for dengue diagnosis as it was the standard test provided by the government due to the high burden of the disease, and routinely used at the study sites, we have also mentioned about this information in the Introduction. Similarly, the EUROIMMUN IgM antibody assay was chosen as the best available method, having been used in several studies  detect antibodies against chikungunya virus. However, we recognize the potential limitations in rapid point of care diagnostic test kits and these limitations have been addressed in the revised manuscript.

Comment 3:

The study presents a detailed analysis of proportions for chikungunya, dengue, coinfections, and unknown etiologies. Data are well-presented in tables and graphs, aiding visual comprehension. I suggest further exploring the geographical distribution of infections and possible associated environmental factors. The section does not sufficiently address the impact of the findings (e.g., implications for hospital management). A deeper analysis of coinfections and their clinical implications would add significant value.

Response: 

Thank you for your comment and suggestion. We would like to clarify that, due to the study design, we were unable to investigate environmental factors, and this limitation is addressed in the revised manuscript.egarding the geographical distribution, we would like to draw the reviewer’s attention to Section 3.4, "Distribution of Proportions of Patients Who Presented to Provincial Hospitals." In this section, we have provided detailed information on the variations in the distribution of cases across the four provincial hospitals.

Comment 4:

The discussion interprets the findings in comparison to existing literature, strengthening the study's credibility. It highlights limitations, such as reliance on serology, and proposes alternatives for future research. However, some topics are treated too generically (e.g., alternative arboviruses for unknown etiologies). I suggest including more specific data or hypotheses. The discussion could also delve deeper into the impact of socioeconomic and cultural factors on the underreporting of chikungunya. Proposals for concrete interventions based on the findings, such as diagnostic or vector control strategies, would be beneficial.

Response:

Thank you for this suggestion. The hypothesis regarding alternative etiologies for cases testing negative for dengue is based on our own experience at the Hospital for Tropical Diseases, a research center and university hospital under the Faculty of Tropical Medicine, Mahidol University. While some of this information is unpublished and cannot be directly cited, we have referenced relevant studies. In the revised manuscript, we have incorporated a sentence addressing this point on lines 287-289.

"In our experience at the Hospital for Tropical Diseases in Bangkok, we have encountered several cases where pateints presenting with acute undifferentiated febrile illness were initially worked up for dengue, only to later test positive for Zika or chikungunya, highlighting the importance of considering other common arboviruses in the differential diagnosis."

 Comment 5:

The conclusion reaffirms the study's relevance, highlighting dengue’s predominance and the emergence of chikungunya as a public health issue. It emphasizes the need for local surveillance and diagnostic improvements. However, it could more explicitly propose immediate actions or guidelines for regional hospitals. The conclusion repeats some discussion points without adding new insights.

Response:

Thank you for this suggestion. We have revised the conclusion in the updated manuscript as follows:

"Dengue remains the predominant arboviral infection in Davao de Oro, affecting nearly half of patients with acute undifferentiated febrile illness. The emergence of  chikungunya, previously underreported, indicates broader circulation of arboviruses in the region. Variations in case distribution across hospitals reflect the need for localized surveillance and improved diagnostics. For clinicians, when faced with acute undifferentiated febrile illness, it is crucial to first exclude dengue using available tests. If dengue is negative, clinicians should consider alternative etiologies such as chikungunya and Zika, which are epidemiologically present in the region."

Comment 6:

The references cover a good range of recent and relevant sources, addressing fundamental topics like diagnosis, clinical impact, and arbovirus epidemiology. I suggest including more local or regional sources to reinforce the geographical specificity of the study.

Response:

Thank you. The references have been updated as suggested.

Round 2

Reviewer 2 Report

Comments and Suggestions for Authors

The authors made an effort to respond to the suggestions shared by the reviewer. It seems to me that the work has improved from a scientific and organizational point of view. At this stage I have no further comments or suggestions. I suggest a final and careful reading of the entire text.

Good luck for the authors' future work. 

Author Response

Comment: 

The authors made an effort to respond to the suggestions shared by the reviewer. It seems to me that the work has improved from a scientific and organizational point of view. At this stage I have no further comments or suggestions. I suggest a final and careful reading of the entire text.

Good luck for the authors' future work.

Response:

Thank you for taking the time to review our manuscript and providing valuable, constructive suggestions that have greatly improved it. We sincerely appreciate your efforts and will carefully review the final version as recommended.